# Failure Detection in Medical Image Classification: A Reality Check and Benchmarking Testbed

**Mélanie Bernhardt**                                    *m.bernhardt21@imperial.ac.uk*
*Imperial College London, UK*

**Fabio De Sousa Ribeiro**                              *f.de-sousa-ribeiro@imperial.ac.uk*
*Imperial College London, UK*

**Ben Glocker**                                          *b.glocker@imperial.ac.uk*
*Imperial College London, UK*

**Reviewed on OpenReview:** *https://openreview.net/forum?id=VBHuLfnOMf*

## Abstract

Failure detection in automated image classification is a critical safeguard for clinical deployment. Detected failure cases can be referred to human assessment, ensuring patient safety in computer-aided clinical decision making. Despite its paramount importance, there is insufficient evidence about the ability of state-of-the-art confidence scoring methods to detect test-time failures of classification models in the context of medical imaging. This paper provides a reality check, establishing the performance of in-domain misclassification detection methods, benchmarking 9 widely used confidence scores on 6 medical imaging datasets with different imaging modalities, in multiclass and binary classification settings. Our experiments show that the problem of failure detection is far from being solved. We found that none of the benchmarked advanced methods proposed in the computer vision and machine learning literature can consistently outperform a simple softmax baseline, demonstrating that improved out-of-distribution detection or model calibration do not necessarily translate to improved in-domain misclassification detection. Our developed testbed facilitates future work in this important area[1].

## 1 Introduction

In recent years, several studies have shown the potential of AI systems for improving clinical workflows (Esteva et al., 2017; Oktay et al., 2020; Faust et al., 2022; Calisto et al., 2022; Zhong et al., 2021; Calisto et al., 2021) both in terms of workload reduction and improved patient outcomes. However, none of these AI-systems are perfect and errors will always be present (Shamshirband et al., 2021; Challen et al., 2019). As such, it is of utmost importance to put in place safeguards to prevent potential adverse consequences arising from those failure cases. To this end, some studies have focused on how to best optimize AI-human collaboration for maximizing patients outcomes via the introduction of interpretability tools, or by using the AI as a second reader as opposed to having a stand-alone computational model (Calisto et al., 2022; 2021; Shamshirband et al., 2021; Calisto et al., 2017). However, it has also been shown that such "human safeguards" are not necessarily fail proof since incorrect model predictions can mislead human readers, even experts (Tschandl et al., 2020). On the other hand, some models are designed to be used as a stand-alone system for example in automated triaging (Kim et al., 2022; Annarumma et al., 2019; Van De Leur et al., 2021; Leibig et al., 2022). In these cases, there is a clear need for additional safeguards such as automated failure detection to prevent the use of incorrect predictions and mitigate clinical risk. For clinical

---

[1]Code available at: **https://github.com/melanibe/failure_detection_benchmark**

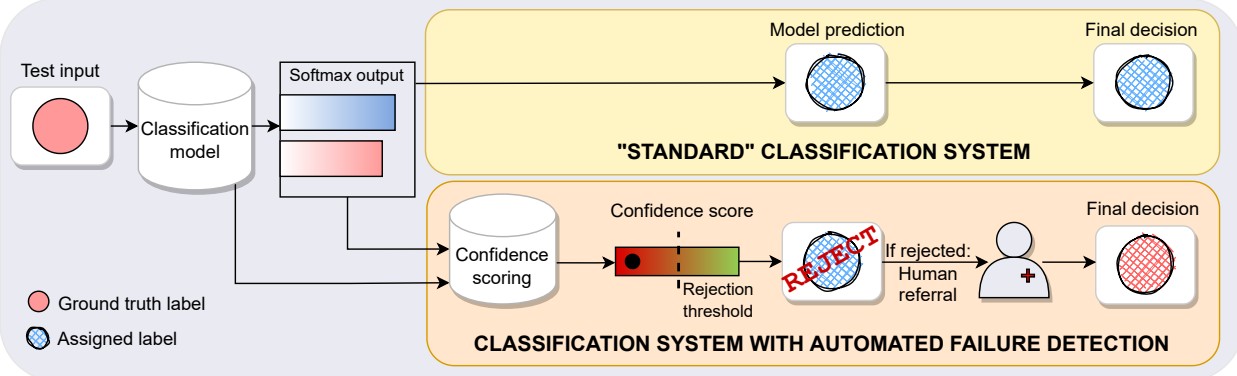

Figure 1: Standard classification system versus a system with automated failure detection. In the latter, a confidence score is computed for each test sample in addition to the model prediction to determine whether the prediction should be accepted or referred for human annotation, based on a chosen rejection threshold.

deployment, reliable failure detection is critical for patient safety, enabling automatic referral to human experts for suspicious predictions (Kompa et al., 2021; Van De Leur et al., 2021; Leibig et al., 2022). For example, some triaging systems include uncertainty-based automatic reject options (Van De Leur et al., 2021; Leibig et al., 2022), i.e. only cases with high prediction confidence are triaged by the AI system.

Despite their increasing use in clinical applications, there is insufficient evidence about the quality of automated *in-domain failure detection* in medical image classification models. This is an important and non-trivial problem, as in most deployment settings the majority of inputs are expected to be in-domain, and no model is error-free, even in the absence of any input perturbation or data corruptions. As depicted in fig. 1, failure detection frameworks are typically divided in two stages: (i) confidence scoring (to quantify the likelihood of the prediction being correct); (ii) a thresholding-step (to reject/refer samples with a low confidence score) (Corbière et al., 2019; Jiang et al., 2018; Band et al., 2021). In such a system, if the confidence scoring scheme is improved, the misclassification detection performance improves.

In the machine learning literature, numerous attempts have been made to obtain better confidence estimates for machine learning classifiers. However, most of the literature in this space has concentrated on benchmarking and improving confidence estimates in terms of robustness to out-of-distribution (OOD) inputs (Hendrycks & Gimpel, 2016; Lakshminarayanan et al., 2016; Gal & Ghahramani, 2016; Daxberger et al., 2021; Maddox et al., 2019), model calibration (Lakshminarayanan et al., 2016; Maddox et al., 2019), or dataset shift (Ovadia et al., 2019). Only few works have studied how useful these state-of-the-art uncertainty estimates and other confidence scores are in the context of *in-domain* failure detection. Recently, Band et al. (2021), extending the work of Leibig et al. (2017), evaluated Bayesian uncertainty estimates in the context of failure detection for in-distribution as well as in data-shifted settings. However, this study has been limited to: (i) the binary diabetic retinopathy detection setting; (ii) focuses exclusively on Bayesian methods. Whereas the Bayesian community has indeed devoted a lot of effort to finding better uncertainty estimates (Lakshminarayanan et al., 2016; Gal & Ghahramani, 2016; Laplace, 1774; Maddox et al., 2019), others have also proposed confidence scores designed explicitly for failure detection, e.g. by analysing the internal representations from the network instead of focusing solely on the output layer (Jiang et al., 2018; Corbière et al., 2019; Van Amersfoort et al., 2020). There is a need for a reality check with a comprehensive comparison of confidence scores for failure detection in terms of methods (Bayesian and non-Bayesian) and, importantly, for a diverse set of medical datasets. This is the purpose of this study.

**Summary of contributions:**

- Our study presents a novel extensive multi-dataset evaluation of *in-domain* failure detection methods in the context of medical imaging classification models. We benchmark 9 different commonly-used uncertainty scores and misclassification detection scores on a diverse set of medical datasets, covering

6 imaging modalities and resolutions ranging from 28×28 to 512×512 pixels, in both multi-class and binary classification settings;

- Surprisingly, we find that none of the benchmarked confidence scores were able to consistently outperform a simple softmax baseline for detecting failures across multiple tasks;

- Our findings demonstrate that reliability against out-of-distribution inputs or improved model calibration does not necessarily translate to improved failure detection for in-domain inputs;

- The developed failure detection testbed (which is made publicly available) aims to facilitate future work, building an essential component for rigorous, comprehensive testing and comparative evaluation of new approaches in the important area of misclassification detection in medical imaging.

## 2 Background

### 2.1 Failure Detection Methods

In this study, we focus on widely used uncertainty scoring schemes, as these would be most likely to be employed in practice at this time. Additionally, we cover a diverse set of methods that we believe are representative of the field, and they can be divided into 5 categories: (i) confidence scores based on softmax outputs (softmax baseline Hendrycks & Gimpel (2016), DOCTOR Granese et al. (2021)); (ii) Bayesian uncertainties (MC-dropout Gal & Ghahramani (2016), Laplace Laplace (1774), SWAG Maddox et al. (2019)); (iii) ensembles (Lakshminarayanan et al., 2016); (iv) non-softmax based models (DUQ, Van Amersfoort et al. (2020)); (v) confidence scores based on feature representations (TrustScore Jiang et al. (2018); ConfidNet Corbière et al. (2019)). A more in depth discussion on the choices of baselines to evaluate can be found in the discussion.

#### 2.1.1 Baseline Confidence Score.

Hendrycks & Gimpel (2016) have shown that the softmax output of neural networks constitutes a good baseline confidence score for failure detection. Precisely, using the softmax output associated to the predicted class as the confidence score i.e. $c(x) = \hat{p}_{\hat{y}}$, where $\hat{p} \in \mathbb{R}$ is the softmax output and $\hat{y}$ is the predicted class index. For multiclass settings (and binary task where the prediction threshold is 0.5) this is equivalent to $c(x) = \max_c \hat{p}_c$.

#### 2.1.2 Uncertainty Estimation.

Recently, several methods have been proposed to obtain better uncertainty/confidence estimates, notably:

- *MC-dropout* (MC): Gal & Ghahramani (2016) showed that training a neural network with dropout regularization (Srivastava et al., 2014) produces a Bayesian approximation of the posterior, where the approximation is obtained by Monte-Carlo sampling of the network's parameters i.e. by applying dropout at test-time and averaging the outputs over several inference passes. The confidence in the prediction can then be approximated by the negative entropy of the outputs; or by taking the softmax confidence score on the averaged outputs.

- *Laplace approximation* (L) (Laplace, 1774): where the posterior is locally approximated with a Gaussian distribution centered at a local maximum (in practice around the maximum-a-posteriori estimate), with covariance matrix corresponding to the local curvature (obtained by an approximation of the Hessian). Recent work from Daxberger et al. (2021) proposed a simple-to-use, lightweight, Python implementation of this approximation, enabling benchmarking the approach easily on pre-trained neural networks.

- *SWAG*: Maddox et al. (2019) define a Gaussian distribution whose mean is parameterized by the stochastic weight averaging (Izmailov et al., 2018) solution, and whose covariance matrix is a low rank matrix plus diagonal covariance derived from the stochastic gradient descent iterates. They then sample several times from this distribution to form the approximate posterior solution.

Table 1: Comparing the technical requirements of each failure detection method considered in this study. 'Base' stands for softmax confidence score baseline (Hendrycks & Gimpel, 2016), 'MC' for Monte-Carlo dropout (Gal & Ghahramani, 2016), '$D_\alpha$' for Doctor (Granese et al., 2021), 'TS' for TrustScore (Jiang et al., 2018), 'L' for Laplace (Laplace, 1774), 'SWAG' (Maddox et al., 2019), DUQ (Van Amersfoort et al., 2020), 'CN' for ConfidNet (Corbière et al., 2019), 'Ens' for Ensembles (Lakshminarayanan et al., 2016).

| | Base | MC | $D_\alpha$ | TS | L | SWAG | DUQ | CN | Ens |
|---|---|---|---|---|---|---|---|---|---|
| Special classification model training | ✗ | ✗ | ✗ | ✗ | ✗ | ✓ | ✓ | ✗ | ✓ |
| Architecture constraints | ✗ | ✓ | ✗ | ✗ | ✗ | ✗ | ✗ | ✗ | ✗ |
| Increased model training cost | ✗ | ✗ | ✗ | ✗ | ✗ | ✓ | ✗ | ✗ | ✓ |
| Post hoc uncertainty score training/fitting | ✗ | ✗ | ✗ | ✓ | ✓ | ✗ | ✗ | ✓ | ✗ |
| Increased inference time | ✗ | ✓ | ✗ | ✓ | ✗ | ✓ | ✗ | ✓ | ✓ |

- *Deep Ensembles* (Ens): Lakshminarayanan et al. (2016) have shown that ensembling predictions from several trained models can yield better calibrated uncertainty estimates than the ones obtain from single deterministic models and even from Bayesian neural networks, in particular due to their increased capacity to capture multi-modal solutions.

- *Deterministic Uncertainty Quantification* (DUQ): Van Amersfoort et al. (2020) estimate predictive confidence based on distances between points and class centroids in the embedding space and demonstrated improved OOD performance.

- *DOCTOR*: Granese et al. (2021) proposed to use $D_\alpha(x) = \frac{1-g(x)}{g(x)}$ where $g(x) = \sum_c \hat{p}_c^2(x)$ as a score quantifying the likelihood of being misclassified (i.e. negative confidence score) as an alternative to the classic predicted softmax confidence score.

These uncertainty estimates have been shown to be state-of-the-art in terms of model calibration and OOD detection (Laplace, 1774; Gal & Ghahramani, 2016; Lakshminarayanan et al., 2016; Maddox et al., 2019; Van Amersfoort et al., 2020). However, only a subset of them has been benchmarked for their failure detection performance and only for a limited number of tasks e.g. Band et al. (2021).

### 2.1.3 Utilizing intermediate representations for confidence scoring.

Another line of research focuses on computing confidence scores using an auxiliary model or intermediate representation, instead of merely looking at the model's final output. Jiang et al. (2018) construct a neighbour-graph in the embedding space from the penultimate layer (on the training set) and use distances in this space to derive *TrustScore* (TS). In practice, this confidence score is defined as the ratio between: (i) the distance between the test point and the closest point that does not belong to the predicted class; (ii) the distance between the test point and closest point that belongs to the predicted class. Corbière et al. (2019) proposed *ConfidNet* (CN) a regression network that is placed on top of the classification model to predict the "true class probability" i.e. the probability predicted by the main model for the true class (as opposed to the probability of the predicted class). In other words, the image is fed through the trained classification model to extract its embedding (penultimate layer), which in turn is fed into ConfidNet which predicts the softmax output for the true class, as originally predicted by the main model. In table 1, we summarize the requirements of each confidence scoring method.

### 2.2 Failure Detection Evaluation

In the recent failure detection literature, a variety of metrics have been used, with seemingly little consensus on which metric is most appropriate. These metrics can be broadly divided in two groups.

### 2.2.1   Classification metrics

The first approach consists of treating the failure detection problem as a classification problem where the goal is to predict whether a given sample has been correctly classified or not, using the confidence score as predictive score. The quality of a given confidence score can then be evaluated with any standard binary classification metric such as ROC-AUC, or the false positive rate (FPR) at a given true positive rate (TPR) (Hendrycks & Gimpel, 2016; Corbière et al., 2019).

### 2.2.2   Selective classification metrics

Another set of metrics derives from the selective classification literature, notably the use of risk-coverage curves (El-Yaniv et al., 2010). This metric consists of plotting the *risk* (typically percentage of errors) as a function of the dataset *coverage* (the percentage of dataset that has not been rejected) (Corbière et al., 2019; El-Yaniv et al., 2010). Others have used alternative versions of this metric such the ROC-AUC-coverage curve (Band et al., 2021) or the risk-score percentiles curve (Lakshminarayanan et al., 2016; Jiang et al., 2018). It is important to note that risk-coverage curves are by definition sensitive to the initial classification performance of the model. Therefore, using this metric to compare various confidence scoring schemes may introduce confounding between model performance and confidence estimation quality. For example, the initial classification accuracy of an ensemble is typically superior to that of a single model, thus we may observe a lower risk-coverage curve for the ensemble even if both models are equally competent at failure detection. This may falsely indicate that one confidence scoring scheme is better than the other. This is the reason why in this work we focus on metrics like ROC-AUC for misclassification detection and FPR at 80% TPR.

### 2.2.3   Note on misclassification detection and model calibration metrics.

In the model uncertainty literature, one common approach to evaluate the ability to capture uncertainty is by measuring model calibration, often quantified via the Expected Calibration Error (ECE) (Naeini et al., 2015; Guo et al., 2017). ECE is obtained by partitioning the prediction into $M$ bins and measuring the difference between the observed accuracy $\text{Acc}(B_m)$ and the average model confidence $\text{Conf}(B_m)$ in each bin $B_m$ that is $\text{ECE} = \sum_m \frac{|B_m|}{n} |\text{Acc}(B_m) - \text{Conf}(B_m)|$. Here, we would like to emphasize that ECE is not necessarily a good metric for measuring how well a model will perform in terms of misclassification detection. Indeed, the ability of a given confidence score to discriminate between correctly and incorrectly classified samples is determined by its ability to correctly attribute lower confidence to misclassified samples than to correctly classified samples. Only the relative ranking of confidences scores affects their ability to find failures, not their absolute value which merely changes the choice of rejection threshold.

To further illustrate why ECE is not necessarily a good metric for failure detection, we can look at the following toy example. Let us suppose that we have two classification models that predict the exact same labels for any given point. "Model 1" is a perfectly calibrated model, that is $\text{Conf}_1(x_i) := \hat{p}_{\hat{y}_i}^{(1)} = P(\mathbb{1}_{y_i = \hat{y}_i})$, where $y_i$ is the true class and $\hat{y}_i$ the predicted class. "Model 2", on the other hand, while keeping the same ranking of its predictions, is overconfident: $\text{Conf}_2(x_i) := \hat{p}_{\hat{y}_i}^{(2)} = 0.9 + 0.1 \cdot P(\mathbb{1}_{y_i = \hat{y}_i})$. We further assume for both models $P(\mathbb{1}_{y_i = \hat{y}_i} | x_i) \sim \text{Bernoulli}(x_i)$, where $x_i \sim \text{Uniform}[0, 1]$. We then simulate the models' responses by sampling 10,000 pairs $(x_i, \mathbb{1}_{y_i = \hat{y}_i})$ according to the above distribution, compute the corresponding confidence scores distribution for Model 1 and Model 2 and compute the ECE as well as the ROC-AUC for misclassification detection. Results are illustrated in fig. 2. After simulation, we obtained a near-perfect ECE of 0.02 for Model 1 but a substantially worse score of 0.45 for Model 2. However, in terms of misclassification performance both models yield the exact same ROC-AUC for misclassification detection, as the predictions of Model 2 are simply a shifted version of the ones of Model 1. That is $\text{ROC-AUC}(\text{Conf}_1(x_i); \mathbb{1}_{y_i = \hat{y}_i}) = \text{ROC-AUC}(\text{Conf}_2(x_i); \mathbb{1}_{y_i = \hat{y}_i}) = .82$.

This simple example provides an intuition of the reason why even though predicted softmax outputs tend to yield over-confident probability estimates (Hendrycks & Gimpel, 2016; Guo et al., 2017), they were still shown to provide a good baseline for misclassification detection (Hendrycks & Gimpel, 2016; Granese et al., 2021). Given the fact that misclassification detection metrics are only sensitive to the relative ranking of predicted probabilities, methods like temperature scaling (Guo et al., 2017) aiming to calibrate outputs by

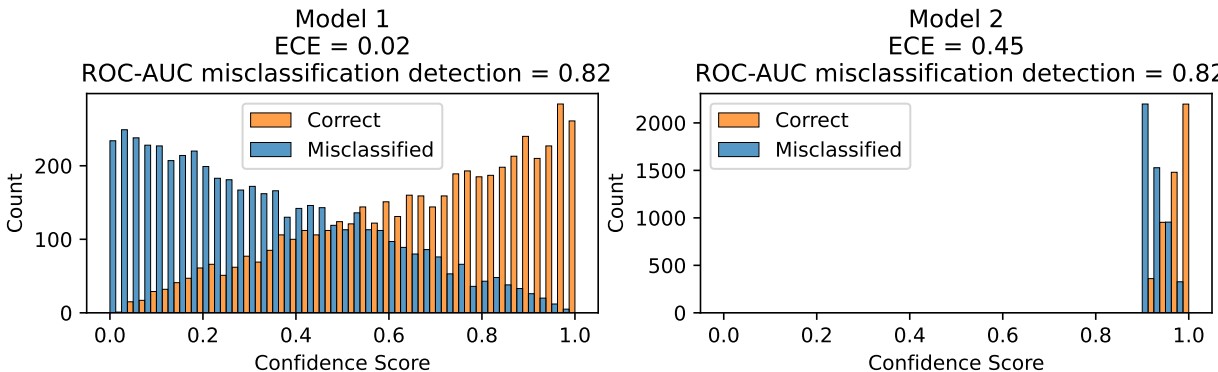

Figure 2: Distribution of observed confidence scores distributions for 'Model 1' and 'Model 2' in the toy example described in section 2.2.3. We can see that Expected Calibration Error (ECE) is not necessarily a good metric for measuring the misclassification detection performance of a given confidence score.

rescaling them, do not impact the failure detection abilities of a given model (but merely change the value of the rejection threshold).

## 3 Experimental Setup

### 3.1 Datasets

First, we evaluate confidence scores on 3 tasks from MedMNIST-v2 (Yang et al., 2021) (all with test set size above 5,000 images). PathMNIST (Kather et al., 2019) consists of non-overlapping patches from histology slides annotated with 9 colon diseases classes (with train and test splits from different clinical centers). TissueMNIST (Ljosa et al., 2012) is comprised of kidney cortex cells microscope images, classified into 8 classes of cell subtypes. OrganAMNIST (Xu et al., 2019) is comprised of center slices from abdominal CT images in axial view, classified by organ type (11 classes). All images have a resolution of $28 \times 28$ pixels and we use the original train-val-test splits. We have included these small resolution datasets as: (i) this resolution is very common in the machine learning literature as it allows for fast prototyping and benchmarking; (ii) these datasets contain tens of thousands of labelled images, including test sets with more than 5,000 images; (iii) despite their low resolutions, even simple models can achieve good performance on their respective tasks (see table 2), indicating that sufficient signal is still present in the downsampled images.

Secondly, we evaluate on three more challenging medical imaging tasks with higher resolution images, using data from the RSNA Pneumonia Detection Challenge (Shih et al., 2019), the Breast Ultrasound Image Dataset (Al-Dhabyani et al., 2020) (BUSI) and the EyePACS[2] Diabetic Retinopathy Detection Challenge dataset. The RSNA Pneumonia Detection Challenge consists of detecting the presence of pneumonia-like opacities on chest X-rays (26,684 scans). In the BUSI dataset, breast ultrasound scans are classified as normal, benign tumors, and malignant tumors on a smaller dataset (780 images in total). For both datasets, we randomly split the data in 70%-10%-20% train-val-test splits. The EyePACS dataset is comprised of high-resolution retina images depicting various stages of diabetic retinopathy. The original labels consisted of a 5-class classification task, here we follow the approach of Band et al. (2021); Leibig et al. (2017) and binarise the task to distinguish 'sight-threatening diabetic retinopathy' (original classes $\{2, 3, 4\}$) and 'non-sight-threatening diabetic retinopathy' (original classes $\{0, 1\}$). This dataset consists of 35,126 training, 10,906 validation and 42,670 test images. These three datasets add another perspective in terms of resolution (images resized to $224 \times 224$ for RNSA and BUSI, $512 \times 512$ for EyePACS), task difficulty, imaging modality and dataset size compared to the datasets from MedMNIST-v2. Visual examples for each dataset can be found in fig. 3.

---

[2]https://www.kaggle.com/c/diabetic-retinopathy-detection

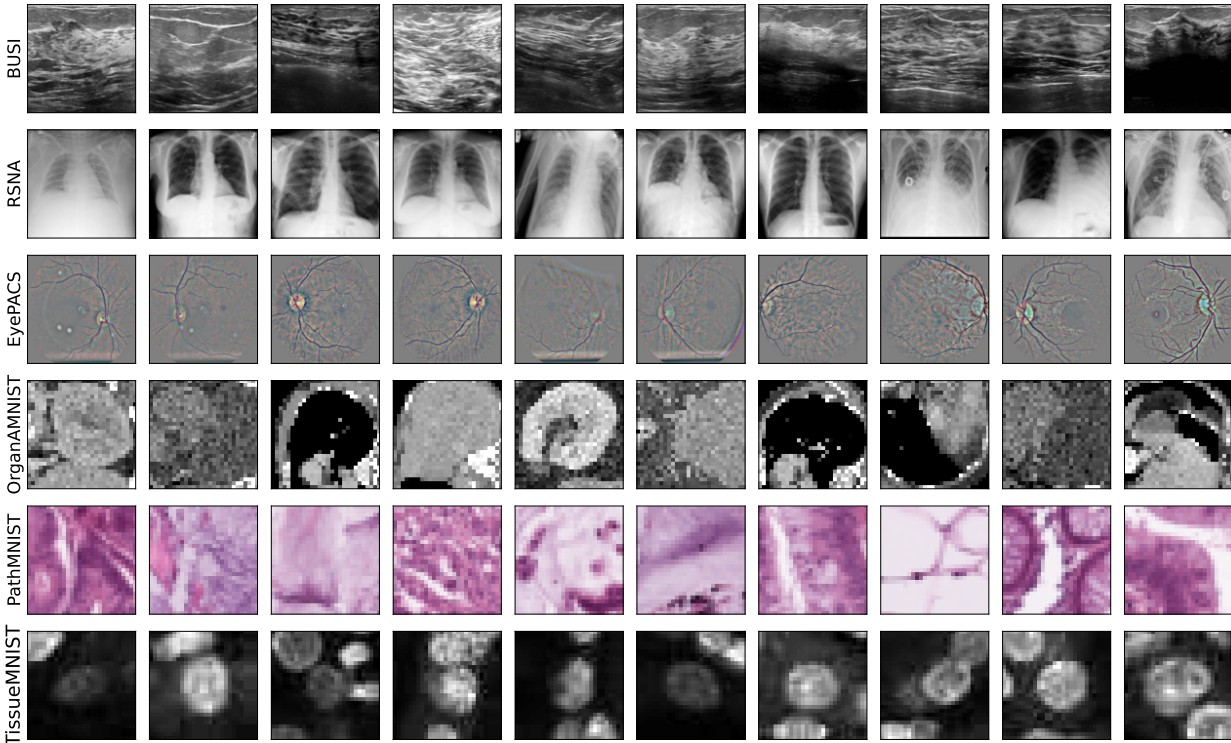

Figure 3: Examples images of each dataset used in the failure detection testbed. For the EyePACS dataset we depict the preprocessed images.

## 3.2 Implementation Details

### 3.2.1 Classification Models.

For our main benchmark we use a ResNet-18 (He et al., 2016) architecture for all the MedMNIST and BUSI tasks and a ResNet-50 model for the RSNA Pneumonia and EyePACS tasks (note that the goal of this study is to assess the suitability of various confidence scoring schemes, not to improve the performance of disease detection models themselves). All models are trained with an additional dropout layer after each weights layer to be able to run the MC-dropout comparison (with dropout probability $p$=0.1 for all experiments, based on validation performance). For all models the learning rate is divided by 10 after 10 epochs with no decrease in validation loss. We stop training after 15 consecutive epochs with no decrease in validation loss and chose the model with the lowest validation loss for testing. For BUSI, all images were resized to 224×224, we applied brightness, contrast, horizontal flips, random rotations and random crop augmentations at training time. For the pneumonia detection task, we used the same augmentations as in Bernhardt et al. (2022). We processed the EyePACS dataset using following Band et al. (2021) which follows the preprocessing from the winning solution of the EyePacs Diabetic Retinopathy Detection Challenge. After preprocessing images were resized to 512×512 prior to input to the network. For both binary tasks (RSNA and EyePACS) the classification threshold was chosen such that the FPR was 20% on the validation set.

### 3.2.2 Failure Detection Methods.

For MC-dropout and SWAG, we set the number of inference passes to 10 (we did not find any notable improvement when increasing the number of inference passes). For SWAG and Ensemble we used the average softmax output of the predicted class as confidence score. For the training of ConfidNet we parameterised network and optimizer following the code provided by Corbière et al. (2019), and also used the checkpoint with the lowest validation AUPR for error detection. We have not further tuned the main model during

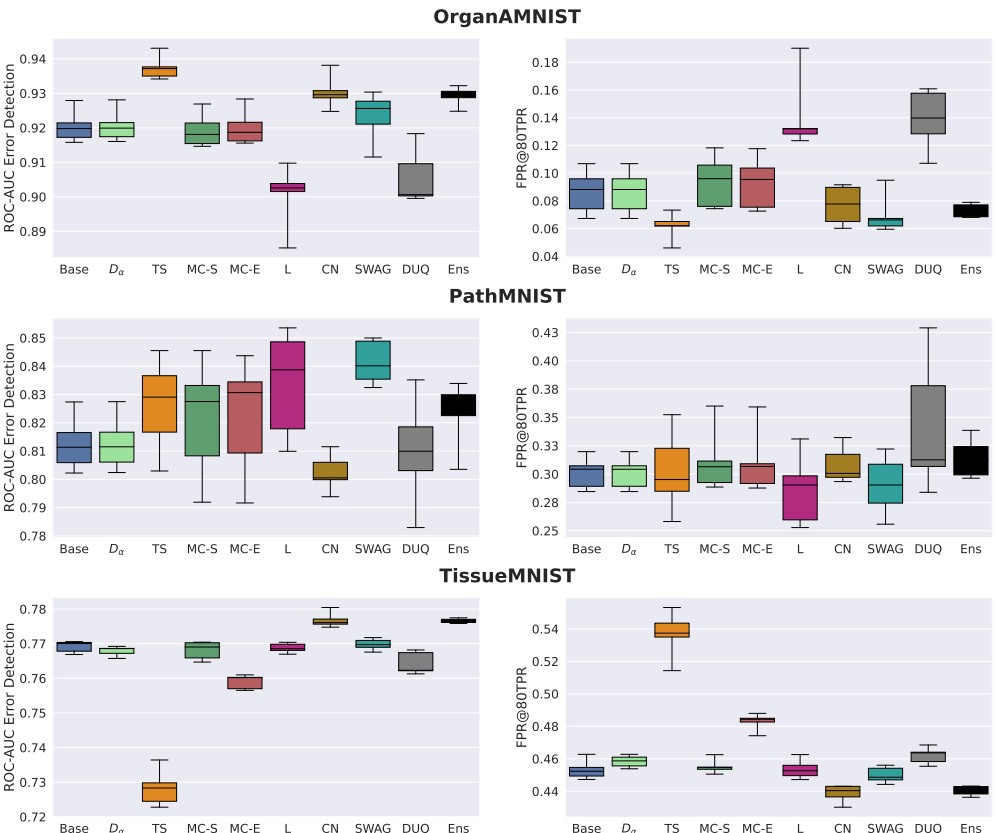

Figure 4: Failure detection benchmark for smaller resolutation datasets: OrganAMNIST, PathMNIST, TissueMNIST. Comparison of Baseline (Base), DOCTOR $D_\alpha$, TrustScore (TS), MC-dropout with softmax score (MC-S), MC-dropout with entropy score (MC-E), Laplace (L), ConfidNet (CN), SWAG, DUQ and Ensemble (Ens). Except for Ensemble, boxplots are constructed with results of repeated training over 5 seeds, whiskers denote minimum and maximum value observed. For Ensemble, we formed 5 different ensembles by taking 5 different combinations of 3 out of the 5 trained models.

the training of ConfidNet. For the Laplace method, we apply the Laplace approximation on the last layer weights using a Kronecker approximation of the Hessian, as per the recommended parameters in Daxberger et al. (2021). For SWAG (Maddox et al., 2019), we tuned the learning rate schedule on the validation set, and for DUQ, we followed Van Amersfoort et al. (2020) for tuning of hyperparameters. Note that when using DUQ for BUSI and EyePACS, we had difficulties finding a set of parameters that provided competitive accuracy with our baseline, nonetheless we show the results obtained with the best hyperparameters found. All the code and hyperparameter configurations to reproduce our results can be found at `https://github.com/melanibe/failure_detection_benchmark`.

## 4 Results

In figs. 4 and 5, we compare the failure detection performance of 9 confidence scores: Baseline (softmax score), TrustScore, Laplace Approximation, MC-dropout, ConfidNet, SWAG, DUQ and Ensemble for 6 different datasets. We evaluate the failure detection performance using *ROC Error Detection*, the ROC-AUC score for classifying correct against misclassified cases (where the positive class is "correctly classified") and *FPR@80TPR*: the FPR at 80% TPR for the error detection task (where the positive class is "correctly classified") i.e. percentage of missed error cases at 20% false alarm rate. We also report classification performance in table 2. To avoid confounding effects between confidence estimation and the model, we use the same architecture and regularization scheme for all confidence scores. In particular, all models were

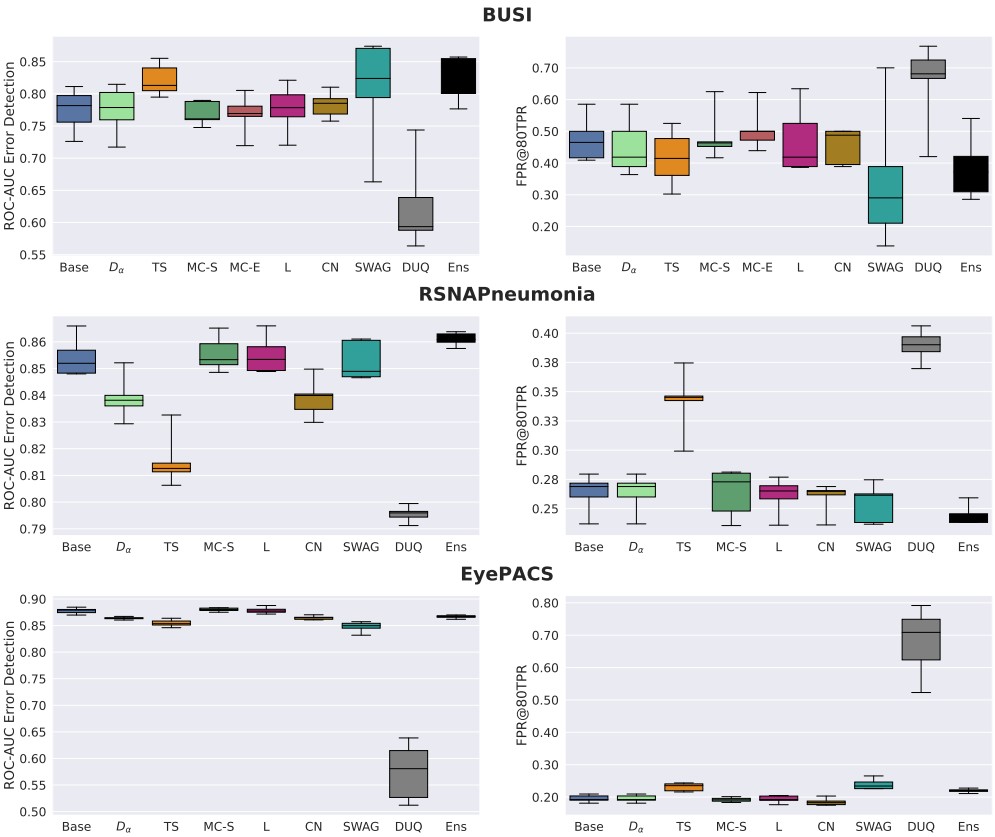

Figure 5: Failure detection benchmark for higher resolution datasets: BUSI, RNSA Pneumonia Detection and EyePACS datasets. Note that we excluded MC-E on the binary tasks as the chosen classification threshold was different from 0.5.

trained with dropout, but we only apply dropout at test-time for the MC-dropout confidence. This differs from other works (Maddox et al., 2019; Band et al., 2021) where numbers reported for the softmax baseline were obtained from a model trained without dropout. We argue that changing the regularisation scheme between different methods can confound the effects of regularisation and uncertainty estimation on failure detection performance, and hence decided to use the same regularization techniques for all models.

The results show that softmax confidence is a strong baseline for misclassification detection. None of the benchmarked methods are able to significantly outperform this baseline consistently across datasets, not even computationally expensive methods such as ConfidNet, MC-dropout, SWAG or Ensembling, and this across all our misclassification detection metrics. In particular, we observe that ensembles increase model performance over a single deterministic model across datasets, however, when evaluating their misclassification detection ability with metrics controlling for classification accuracy, they do not improve consistently over a simple baseline. To test this hypothesis further, we reran all experiments using two additional model architectures (DenseNet-121 and WideResnet-50) and observe the same phenomenon: none of the methods outperform the baseline softmax score, see fig. 6 where we report failure detection ROC-AUC for each model, dataset combination. Moreover, we observed that TrustScore is significantly underperforming for TissueM-NIST and RSNA Pneumonia, whereas it is performing well on other datasets. To understand why, in fig. 7 we analysed the t-SNE (Van der Maaten & Hinton, 2008) representation of the embedding spaces and we observed that this correlates with a less well class-separated embedding space.

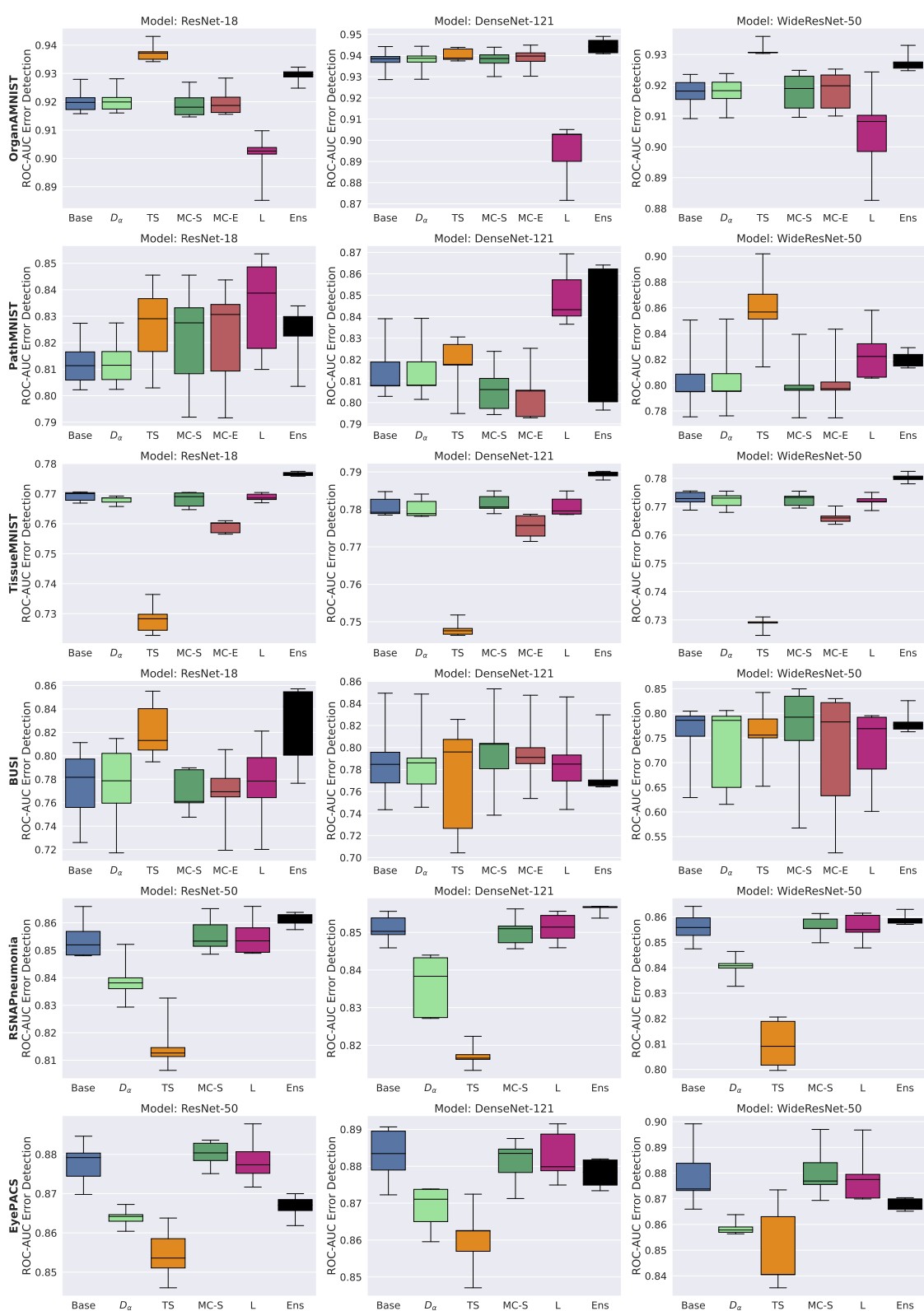

Figure 6: Model architecture effect ablation study. SWAG, ConfidNet and DUQ not included in this ablation because of computational cost (as they all require separate training).

Table 2: Classification performance for ResNet models: accuracy for multiclass tasks and ROC-AUC for binary tasks. For all methods except Ensemble, we report the average over 5 seeds, standard deviation in brackets. For Ensemble, we formed 5 different ensembles by taking 5 different combinations of 3 out of the 5 trained models, we report the average over of the 5 ensembles, standard deviation in brackets.

| Dataset | Baseline | MC | Laplace | SWAG | DUQ | Ensemble |
|---|---|---|---|---|---|---|
| OrganAMNIST | .902 (.005) | .902 (.005) | .901 (.006) | .910 (.004) | .917 (.004) | **.921 (.002)** |
| PathMNIST | .838 (.009) | .832 (.010) | .835 (.010) | .830 (.010) | .828 (.025) | **.853 (.002)** |
| TissueMNIST | .663 (.004) | .664 (.003) | .663 (.004) | .670 (.001) | .655 (.007) | **.682 (.001)** |
| BUSI | .740 (.020) | .740 (.021) | .740 (.020) | **.792 (.017)** | .561 (.000) | .742 (.017) |
| RSNA | .871 (.007) | .873 (.006) | .871 (.006) | .873 (.003) | .865 (.003) | **.877 (.003)** |
| EyePACS | .899 (.006) | .899 (.007) | .899 (.007) | .913 (.004) | .730 (.007) | **.918(.002)** |

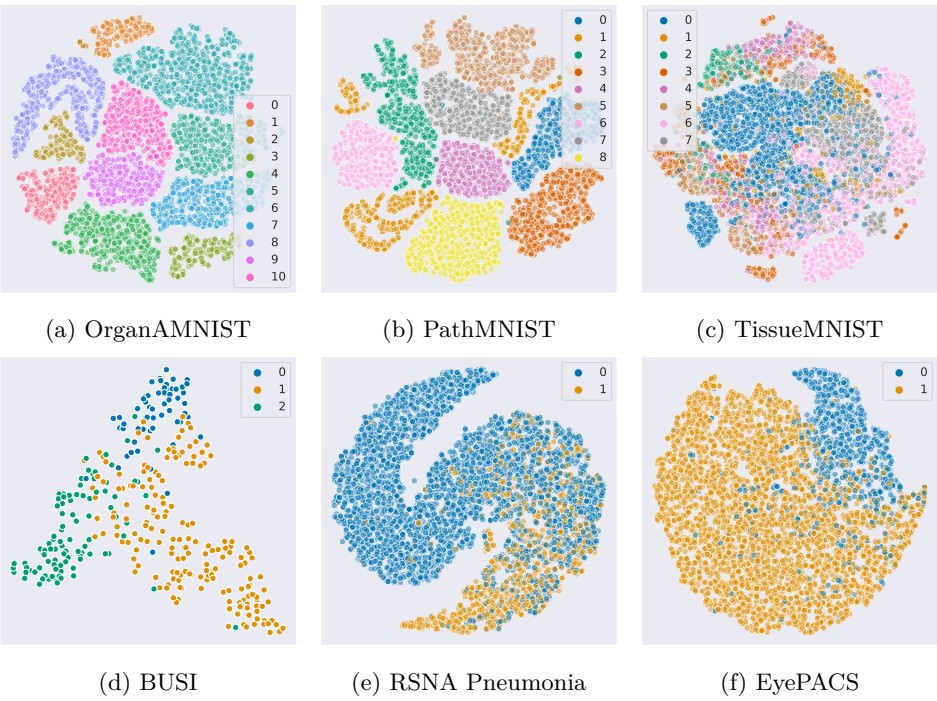

(a) OrganAMNIST    (b) PathMNIST    (c) TissueMNIST

(d) BUSI    (e) RSNA Pneumonia    (f) EyePACS

Figure 7: t-SNE representation of the embedding space on the training set for each dataset for ResNet models. Note that for better visibility, a maximum of 5,000 random samples are plotted (but we used the entire training set to construct the t-SNE representation).

## 5  Discussion

**Key take-aways.**  Our investigation showed that the softmax confidence score baseline is difficult to beat: none of the benchmarked advanced confidence scoring methods were able to consistently outperform the baseline across datasets. Crucially, our experiments show that previously demonstrated improved robustness for out-of-distribution detection or model calibration does not necessarily translate to improvements in error detection for in-domain inputs. This finding indicates that the in-domain failure detection is an orthogonal, complementary task to these related tasks and should as such be evaluated separately. This task has been vastly understudied in comparison to the attention that OOD detection has received and we hope that these findings will foster more research in this key area. Moreover, our study distinguishes itself by the particular care taken to avoid confounding effects between model performance and improvements in confidence score quality, both by: (i) our choice of evaluation metrics; (ii) whenever possible, keeping the same classification model to evaluate various confidence scores, and ensuring that regularization schemes such as dropout were kept constant for all models evaluated.

**Practical implications.**  Our study presents the first extensive failure detection benchmark covering a wide range of medical datasets with various image resolutions (from $28 \times 28$ to $512 \times 512$ pixels), a wide-range of imaging modalities and with tasks of varying difficulty. In particular, our work shows that care should be taken prior to deploying an automated failure detection system to a real-world scenario by carefully evaluating the performance of such a system on the practical task at hand. Our results indicate that current uncertainty scoring, including a simple softmax baseline, are able to detect some misclassified cases at a substantially better than chance level (ROCAUC $> .75$ for all benchmarked datasets), demonstrating the potential to improve practical performance of AI systems by deferring suspicious cases to humans. It is important to note that regardless of the dataset or the confidence score, there is still a clear margin for improvement as FPR@80TPR (the proportion of missed errors at a reasonable false alarm rate of 20%) remain very high for most datasets. We highlight that our work is not meant as a one-time assessment, but as a starting point towards a more systematic and objective evaluation of failure detection methods. We envision this testbed to dynamically grow, inviting more results to be added over time through contributions from the community.

**Baseline choices and limitations.**  As mentioned in the background section, in this study we focused on most widely used confidence scoring schemes, as these are most likely to be deployed in practical scenarios and are thus at a higher risk of being misused. Nevertheless, despite our best efforts to choose representative methods of the field – covering a diverse set of approaches to confidence scoring (Bayesian uncertainties, softmax-based scores, scores based on intermediate representations) – this benchmark is naturally not an exhaustive evaluation of all uncertainty and confidence scores ever presented in the literature. As such, we make no claims as to whether other uncertainty or confidence score methods proposed in literature are (or will be) able to beat the softmax baseline consistently. Nonetheless, our study clearly demonstrates that failure detection ought to be carefully studied in isolation, as other tasks such as OOD or calibration do not provide a reliable proxy for the misclassification detection performance of a given confidence score. Moreover, given the variability of the results from one dataset to another, this experimental setup demonstrates the need for more broad evaluation of new confidence scores in the future.

## 6  Conclusion

In this study, we conducted a thorough evaluation of 9 widely used confidence scores for failure detection on 6 medical datasets. We found that among all the benchmarked confidence scores, none were able to outperform the softmax baseline consistently across datasets – demonstrating that improved model calibration or performance on OOD does not necessarily translate to improvement for in-domain misclassification detection. Overall, our study strongly suggests that failure detection requires further research towards finding more appropriate confidence scoring schemes, as this topic is a key concern for the medical imaging community where safety concerns are paramount for deployment of AI models. This work facilitates further investigations by providing a complete testbed for evaluating future progress in this space in a rigorous and comprehensive manner.

## Broader Impact Statement

This work evaluates failure detection methods in the context of medical imaging and is therefore closely related to AI safety concerns for healthcare-related deployment. This work provides a testbed to facilitate further work in the key area of automated failure detection and calls upon more research in this direction. Nevertheless, this work does not argue for the use of any particular method in practice and this testbed has been designed for research purposes; as such any conclusions obtained using this testbed should always be validated locally using representative data for the considered practical deployment scenario.

## Acknowledgments

M.B. is funded by an Imperial College London President's scholarship. This project has received funding from the European Research Council (ERC) under the European Union's Horizon 2020 research and innovation programme (Grant Agreement No. 757173, Project MIRA).

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
