# OpenReview forum: "Failure Detection in Medical Image Classification: A Reality Check and Benchmarking Testbed"
_TMLR — Accepted by TMLR_

### Review · Reviewer_K7ez · 2022-07-06

**Summary Of Contributions:**

In this manuscript, the authors report a study for the evaluation of nine different confidence scores for failure detection on six medical datasets. Specifically, their study presents the first extensive failure detection benchmark covering a wide range of medical datasets with various image resolutions, a wide-range of imaging modalities, and with tasks of varying difficulty. In the end, the authors are suggesting that failure detection requires further research towards finding more appropriate confidence scoreing schemes. Which is not a fair conclusion after all. Although this work is facilitating further investigations, the work findings for the community and final contribution are not clear. In fact, providing a complete testbed for evaluating future progress in this space seems a small contribution for this venue. However, the authors are following a well represented method and metrics to denote their achievements. Overall, this is an exciting line of work, but the current draft of this manuscript has some minor concerns that must be addressed first before publication. Contributions, if so, must be described clearly. Additionally, the findings are also important attributes of such important research work. As follows, further comments are detailing improvements for the next iterations of the manuscript.

**Broader Impact Concerns:**

Although this is an interesting read and its topic is timely, as is, the paper has some minor concerns, for the impact and its implications, that must be addressed first. That said, the authors need to improve the current status of the manuscript.

**Requested Changes:**

There is much to appreciate in this manuscript, as it investigates that the softmax confidence score baseline is difficult to beat. However, the provided findings and work implications are not enough, and the discussion section would need to be considerably strengthened by a more robust engagement with the cited literature. I would encourage the authors to think about how their findings extend or add to the ML and medical literature in this space in order to refine their contribution to these fields. For instance, it would be interesting to understand the application of these failure detection techniques to a real scenario [1, 4]. At least, a small discussion on how could these techniques help real problems, such as different medical domains [2, 3, 5].

The paper needs a brief proofreading as some English phrases seem to be off. Be aware of tenses also. It would be wise to proofread the paper and correct these, as well as other spelling errors.

The related work seems scarce for a work around the healthcare domain. The paper must discuss recent work in a methodological sense and base their work on the proposed approach. A lot of work is already available into this topic [1, 3, 5]. Specifically, the scientific community has numerous works on this topic [1, 5], but authors could also map some literature around other domains [3, 5], where these level of approaches are essential.

To conclude, the paper is almost ready for publication and the authors did a splendid work. However, I strongly recommend the authors improve the manuscript for the next iterations of the work, as this is a chief importance domain.

# Major Comments

1. Contributions and impact for this scientific community are not clear;

2. The novelty of the work is also dubious;

3. Findings and implications of the work are merely reported;

# Minor Comments

4. The literature must be further addressed;

5. The paper hardly fails on missing citations;

6. Lack of necessary context about the domain and problem;

7. Discussion section could be improved;

8. It would be wise to proofread the paper, as well as correct grammar and other spelling errors;

#### Missing References

[1] https://doi.org/10.1016/j.compbiomed.2022.105407

[2] https://doi.org/10.1016/j.artmed.2022.102285

[3] https://doi.org/10.1109/EMBC46164.2021.9630228

[4] https://doi.org/10.1016/j.ijhcs.2021.102607

[5] https://doi.org/10.1016/j.jbi.2020.103627

[6] https://doi.org/10.1145/3399715.3399744

[7] https://doi.org/10.1016/j.cmpb.2019.03.008

[8] https://doi.org/10.1145/3132272.3134111

**Strengths And Weaknesses:**

#### Strengths

**1.1.** The manuscript addresses key concerns of failure detection in healthcare;

**1.2.** The manuscript brings forward novel perspectives for ML developments in healthcare;

#### Weaknesses

**2.1.** Contributions and impact for this scientific community are not clear;

**2.2.** The novelty of the work is also dubious;

**2.3.** Findings and implications of the work are merely discussed;

---

> ### Author Response · Authors · 2022-07-13
> **Authors' response**
>
> We would like to thank the reviewer for the feedback and improvements suggestions for our manuscript.
>
> - As suggested by the reviewer, we have added more background, references and context of our work from a healthcare point of view. _Please see our substantially extended introduction (changes in blue in the pdf)_, in which we included more concrete examples of medical imaging applications involving AI, studies focused on AI-Human collaboration (as suggested) as well as existing systems incorporating failure detection methods. We hope that this gives the necessary context to our work. We thank the reviewer for the suggestion and provided references.
> - As requested, we have also added a paragraph discussing the practical implications of our work, _see newly added “practical implications paragraph” in the discussion_.
> - In order to further strengthen our take-away message, we have clearly outlined our contributions in a separate paragraph at the end of the introduction (_see “contributions summary”_). Indeed, as pointed out by the reviewer this work provides the first extensive multi-dataset study of failure detection techniques in the context of misclassification detection for medical imaging tasks, addressing - as stated by the reviewer - one “key concern of failure detection in healthcare”. As such this study provides insights both for the medical imaging community where safety concerns are paramount for deployment of AI models; but also provides valuable insights for the ML community as our study demonstrates that improved model calibration or performance on OOD does not necessarily translate to improvement for in-domain misclassification detection. This finding indicates that failure detection needs to be treated as an orthogonal, complementary task when evaluating the potential of a new uncertainty score (in other words as stated by reviewer XowJ: “Just because confidence scores are useful for OOD detection, we should not assume they will be useful for ID failure detection”).
> - Additionally the updated version of the manuscript now contains the _requested Broader Impact Statement_.

---

### Review · Reviewer_vyW2 · 2022-07-06

**Summary Of Contributions:**


This paper argues that "previously demonstrated improved robustness for out-of-distribution detection or model calibration does not necessarily translate to improvements in error detection for in- domain inputs".

They look at 6 different methods across a variety of small and large resolution datasets.


**Requested Changes:**

Clarify the scope. Narrow it to the specific point you are trying to make. Then justify the selection of methods and datasets to address this specific point. If you want to make a broad sweeping statement of all failure detection methods then you need to have a good argument that me evaluating some random method I find won't change the conclusion of your paper.

Enumerate the insights that will help move the field forward.


**Strengths And Weaknesses:**

The paper starts with a focus on out-of-distribution detection methods and then focuses on a few scattered methods related to uncertainty estimates. There is a significant number of methods focused on just out of distribution methods. Such as a comprehensive OOD benchmark here: https://arxiv.org/abs/1809.04729 and then one specifically for Medical OOD here: https://arxiv.org/abs/2007.04250 Both of these works explore many OOD methods not used in this submission. It is impossible to evaluate all methods but there needs to be some argument for the methods selected which relate to the point that is being made.

Because of this I think the scope of the paper needs to be focused on uncertainty estimates and with that I do not think it is comprehensive. So some argument should be given for why the methods chosen are representative. There are 24 papers with uncertainty in the title in NeurIPS 2021 alone.

I do not see this work providing actionable insights. There is no focus on modality, types of features, or types of tasks. What insights can I take away from this work to better develop a method for uncertainty estimation?

The author states "our study strongly suggests that failure detection requires further research towards finding more appropriate confidence scoring schemes" which I don't believe is surprising and I don't think this work is comprehensive enough to prove it either.

---

> ### Author Response · Authors · 2022-07-13
> **Authors' response**
>
> We would like to thank the reviewer for the time to review our work and the constructive feedback. Herein, we would like to provide some clarifications regarding some of the concerns raised by the reviewer.
> - Importance of the problem: this work focuses on in-domain misclassification detection, which is a key concern in sensitive AI applications such as healthcare. For example, recently AI several systems have been proposed for automated triaging (e.g. for [Breast Screening](https://www.thelancet.com/journals/landig/article/PIIS2589-7500(22)00070-X/fulltext)) some of which have already been CE marked and ready to be deployed (see for example [this blog post](https://laurenoakdenrayner.com/2022/07/04/no-doctor-required-autonomy-anomalies-and-magic-puddings/) for a discussion). As these systems get progressively deployed and start impacting the patient’s care, it is necessary to put in place safeguards to detect bad predictions in order to automatically refer these cases to a human reader instead. This is where automated failure detection plays an important role. Despite its importance, this topic is currently far less studied than other uncertainty related tasks such as out-of-distribution detection or model calibration (as the review mentions there are many works focusing on these tasks). _We have clarified the importance of the problem in the updated version of our introduction, (changes in blue in the updated pdf)_.
> - Scope of the work: we would like here to re-highlight the scope of this work. First of all, this work focuses on in-domain failure detection which is an orthogonal, complementary problem to out-of-distribution detection. Whereas out-of-distribution detection focuses on detecting invalid inputs (or inputs outside of the training distribution), in-domain failure detection focuses on detecting samples that have been misclassified despite coming from the same distribution as the training set. This is a non-trivial problem and of high importance as in most medical setting the majority of inputs are expected to be in-domain inputs (as pointed out by reviewer XowJ :”This is problematic, as ID is the more commonly occurring and thus more practical use case in the clinical setting.”) _This has been clarified in the new version introduction and in the new contributions statement_.
> - Choice of baselines: as pointed out by the reviewer “It is impossible to evaluate all methods”. For that reason in this work we have focused on widely used and benchmarked uncertainty scoring schemes, as these would be most likely to be considered in practice at this time. For representativeness, we have taken care of covering a diverse set of methods that can be divided into 5 categories: Bayesian uncertainties (MC-dropout, Laplace, SWAG), Ensembles, confidence scores based on feature representations (TrustScore, ConfidNet), non-softmax based models (DUQ), confidence scores based on softmax outputs (softmax baseline, DOCTOR).  Here we do not claim that no uncertainty or confidence score proposed in the literature is / will be able to beat the softmax baseline. However, our study clearly demonstrates that failure detection needs to be studied on its own, within a well designed and dedicated testbed, as other tasks such as OOD or calibration do not provide a reliable proxy for the misclassification detection performance of a given confidence score. _To clarify this point in our updated manuscript we have now added a “choice of confidence scores and limitations” paragraph in the discussion section_.
> - Summary of contributions: this work provides the first extensive multi-dataset study of failure detection techniques in the context of misclassification detection for medical imaging tasks. This study demonstrates that improved model calibration or performance on OOD does not necessarily translate to improvement for in-domain misclassification detection, as none of the benchmarked methods was able to outperform as simple softmax baseline consistently across several datasets. This finding shows that failure detection needs to be treated as an orthogonal task when evaluating the potential of a new uncertainty score (or as stated by reviewer XowJ: “Just because confidence scores are useful for OOD detection, we should not assume they will be useful for ID failure detection”). Finally, the diversity of results across datasets highlights the importance of having a comprehensive testbed for evaluating failure detection methods in the future. Our work is not meant as a one-time assessment, but as a starting point towards a more systematic and objective evaluation of failure detection methods. We envision this testbed to dynamically grow, inviting more results to be added over time through contributions from the community. _This is highlighted in the updated contributions paragraph in the manuscript and newly added practical implications paragraph in the discussion section_.

---

### Review · Reviewer_XowJ · 2022-07-07

**Summary Of Contributions:**

The authors present an empirical study comparing confidence scores for in-distribution (ID) failure detection across Bayesian and Non-Bayesian methods, several medical imaging datasets, and multiple deep learning architectures. Their main contribution is to show that no confidence score outperforms a simple softmax baseline consistently across datasets. While these methods may be useful for out-of-distribution (OOD) detection, this does not necessarily mean they are useful for ID failure detection. This is problematic, as ID is the more commonly occurring and thus more practical use case in the clinical setting.

**Broader Impact Concerns:**

There is no broader impact statement present, and one does not seem necessary. The paper addresses ID failure detection in the medical setting, where improvements are likely to benefit society as a whole without harming any particular demographic.

**Requested Changes:**

The paper would be stronger with the incorporation of at least one more publicly available large-scale medical imaging dataset such as CheXpert.

**Strengths And Weaknesses:**

The paper is well-written and well-executed, and has a clear and important take-away message. Just because confidence scores are useful for OOD detection, we should not assume they will be useful for ID failure detection. That none of the methods consistently outperformed the softmax baseline is concerning, and points to a need to develop better methods for this important use case in medical AI. The experiments included a large variety of confidence scoring methods, and the choice of model architectures was sound.

The main weakness of the paper is that out of six datasets, three are trivial with 28x28 image resolution, which does not reflect a realistic medical AI setting. Additionally, of the three remaining datasets, the breast ultrasound dataset only has 780 images. Since there are only two datasets with tens of thousands of high-resolution images, I am not certain whether the paper's conclusions would transfer to a state-of-the-art medical AI setting with hundreds of thousands or millions of high-resolution images.

---

> ### Author Response · Authors · 2022-07-13
> **Authors' response**
>
> We are most appreciative for the thorough review and we would like to thank the reviewer for the valuable suggestions. We are delighted to see that the reviewer enjoyed our work and appreciated the value of our work. In the following, we would like to respond  to one of the concerns raised by the reviewer regarding the choice of datasets in terms of scale and resolutions:
>
> - First of all, we would like to thank the reviewer for the suggested addition to our testbed. We had actually considered using CheXpert (or related NIH dataset) for this application, and we had carefully investigated their utility. However, these datasets are all labeled by an automated NLP-based labeling tool. And as shown in the CheXpert paper itself, these labels can contain high levels of noise (e.g. AUC of only .78 for “no finding” label against human annotation). This is particularly problematic for a study like ours that focuses on analyzing model errors. Some important issues with CheXpert and similar datasets were recently discussed in this brief analysis in Nature Medicine ([https://doi.org/10.1038/s41591-022-01846-8](https://doi.org/10.1038/s41591-022-01846-8)). This is the reason why chose to use the RSNA Pneumonia Detection task instead as a representative for a high-resolution chest X-ray dataset. Labels in this dataset have been reviewed by multiple radiologists and adjudicated in case of disagreement, ensuring a high label quality. Moreover, the resolution of the images in the RSNA dataset is the same as CheXpert and Pneumonia Detection is actually one of the tasks in CheXpert. Finally, the dataset contains more than 25,000 images which is very large for a manually labeled medical dataset. We hope that this reasoning sheds light on our dataset choice and that it makes sense to the reviewer why we would argue not to include CheXpert. We would be happy to discuss this further if the reviewer feels this has not been answered sufficiently.
> - Secondly, the reason why we chose to include small-resolution datasets from MedMNIST were 3-fold: (i) this resolution is a very common resolution in the ML litterature (same resolution as MNIST, CIFAR) as it allows for fast prototyping and iterations when developing new methods (ii) these datasets are extremely big labeled datasets (which is rare in the healthcare domain), in particular we here focused on the datasets with the biggest test sets (> 5,000 images) for reliable evaluation, (iii) despite their small resolutions even simple models like resnet achieve good performance on the tasks at hand, indicating a sufficient signal in the images. _We added this justification in the updated manuscript in the datasets section (subsection 3.1, changes in blue)_.
> - Thirdly, we absolutely agree with the reviewer that the Breast dataset is very small. However, we still chose to include this dataset as this is a very common characteristic in medical imaging applications and a representative example of the real-world. We found it would be interesting to include results obtained on smaller scale datasets to compare with other datasets that have tens of thousands of images. We aimed to address concerns of stability of results (due to the small test set) by including results of multiple training runs and representing the spread with the boxplots.
>
> Finally, as suggested by the reviewer we have _added a Broader Impact Statement_ at the end of the updated manuscript.

---

### Decision · Action_Editors · 2022-09-26

**Recommendation:** Accept with minor revision

**Comment:**

First of all, I would like to apologize for the delay in making a decision.

After carefully reading the paper, the reviews and the responses, I think that although the idea in this paper is not particularly exciting (could an empirical analysis of existing methods be particularly exciting?), it is a well executed empirical work on a relevant research question with practical implications.

The reviews for this paper were of high quality and the reviewers brought a number of valid concerns. Especially, some points brought up by the reviewer vyW2 might not have been sufficiently addressed (e.g. the rationale for the choice of methods, I recommend to acknowledge this in the final version of this paper). Overall, with the vote 2:1 for rejection after rebuttal indicates that the authors did not manage to fully convince the reviewers. Having said that, I think the reviewers of this paper might have been on the harsher side in terms of their final recommendations.

I think this is a valid contribution, with correctly executed experiments, which will be of interested to the community. Given that "TMLR emphasizes technical correctness over subjective significance", I recommend that this paper to be accepted with minor revisions. The main effort should be to incorporate the discussion points brought up by the reviewer yvW2 to the best of your ability. Finally, I think it would be beneficial to carefully proofread this paper by some of the experienced co-authors to make the paper flow a little better (e.g. in the background section).

---

> ### Author Response · Authors · 2022-10-21
> **Camera-ready version**
>
> We thank the editor for the useful feedback and for recognising the value of our work. The uploaded camera-ready version now reflects the requested changes notably regarding choices of methods (see discussion section, baselines choices and limitations as well as background section) extending the discussion section added during the rebuttal phase. We thank the reviewers and the action editor for their time in the review process.